# Mixed neural posterior estimation for simulators with discrete and continuous parameters

## Abstract

Neural Posterior Estimation (NPE) enables rapid parameter inference for complex simulators with intractable likelihoods. NPE trains an inference network to estimate a probability density over parameters given data, typically assumed to be *continuous*. However, many scientific models involve parameter spaces that are *mixed*, that is, they contain both discrete and continuous dimensions. We address this limitation by extending NPE to mixed parameter spaces through an inference network that jointly handles discrete and continuous parameters. The inference network factorizes the joint posterior into discrete and continuous components, combining an autoregressive classifier for the discrete parameters with a generative model for the continuous parameters, trained jointly under a single simulation-based objective. In addition, we propose a diagnostic tool to assess the calibration of the mixed posterior approximation. Across tractable toy examples and real-world scientific simulators, our joint inference approach yields accurate and calibrated posteriors.

## 1 Introduction

Neural Posterior Estimation (NPE) has become a central tool for simulation-based inference (SBI) (Cranmer et al., 2020; Deistler et al., 2025a). NPE enables posterior inference in simulator models using only simulations from the forward model, without requiring access to likelihood evaluations. This setting arises across the natural sciences, cognitive science, epidemiology, and engineering (McKinley et al., 2014; Gonçalves et al., 2020; Fengler et al., 2021). In addition, NPE amortizes the cost of simulation and training and rapidly performs inference for any observation, enabling real-time and high-throughput applications (Dax et al., 2021; von Krause et al., 2022). Over the past years, NPE has seen broad adoption, with software libraries making it accessible to domain scientists with little tuning or customization needed (Boelts et al., 2025; Kühmichel et al., 2026). As these libraries lower the barrier to adoption, NPE is increasingly applied directly to simulators as they arise in practice.

A fundamental challenge in using NPE is that many simulators involve both continuous and discrete parameters (i.e., categorical variables). We refer to these simulators as mixed (discrete–continuous). Discrete parameters often arise naturally: phenomena such as ion channel type in neuroscience (Schröder & Macke, 2024), change-point locations in time-series simulators (Adams & MacKay, 2007; Altamirano et al., 2023), switching dynamical systems (Linderman et al., 2017; Fu et al., 2024), queueing simulators with discrete service regimes (Gross et al., 2008), or embedded model selection variables (Radev et al., 2021; Gloeckler et al., 2026) are inherently discrete. Handling such mixed parameter spaces is non-trivial for NPE. The joint posterior over mixed parameters is a hybrid object: a mixture of continuous densities, one per discrete configuration. Standard continuous inference networks based on normalizing flows or diffusion-based approaches—as commonly used in NPE—cannot naturally represent this. Even if simulator likelihoods are available and standard inference methods like MCMC can be used for mixed parameters, those need to be carefully tuned and they do not benefit from amortization across observations. Thus, despite their ubiquity, simulators with mixed parameters remain insufficiently supported by current SBI methods and toolboxes. Recent work has begun to address this gap from different angles: diffusion-based SBI methods embed discrete parameters in continuous spaces or treat them as model indices (Ghiglino et al., 2026; Schröder & Macke,

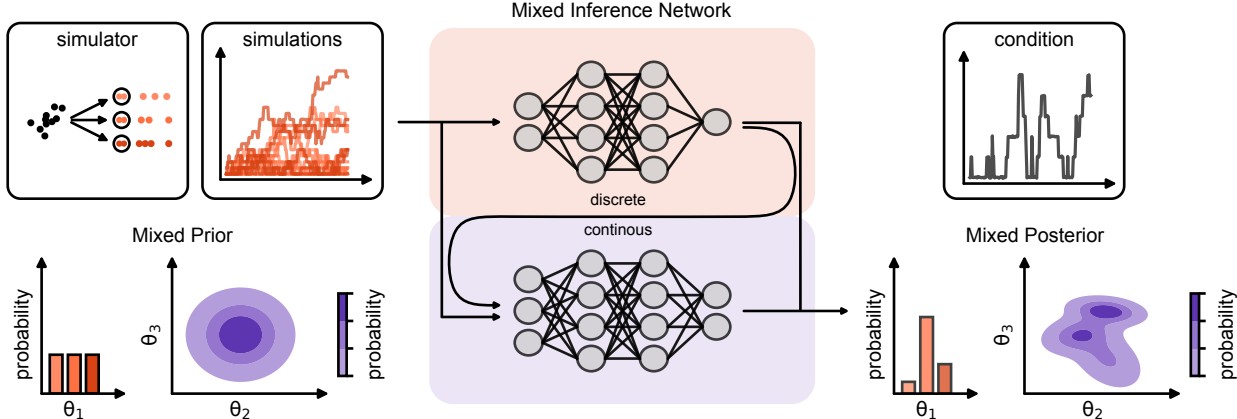

Figure 1: **Inference method overview.** MNPE is trained on a dataset of parameter–simulation pairs $\{(\boldsymbol{\theta}, \boldsymbol{x})\}$ where $\boldsymbol{\theta}$ is sampled from a mixed prior over discrete and continuous parameters. The inference network consists of a subnetwork for the discrete dimensions (orange), implemented as a masked autoregressive density estimator (MADE), and one for the continuous dimensions (violet), which can be a standard inference network for continuous posterior estimation (e.g., normalizing flow, diffusion model). At inference time, the network is conditioned on an observation $\boldsymbol{x}_o$ and provides the joint posterior over discrete and continuous dimensions.

2024; Gloeckler et al., 2026), while mixed MCMC approaches extend Hamiltonian dynamics to handle discrete variables alongside continuous ones (Nishimura et al., 2020; Zhou, 2020). We discuss these approaches and their trade-offs in detail in Section 5.

Here, we extend NPE to mixed parameter spaces by introducing an inference network that handles discrete and continuous parameters jointly (Figure 1). We refer to this extension as Mixed Neural Posterior Estimation (MNPE). Our inference network handles a flexible number of continuous and discrete parameters with arbitrary class counts, and is compatible with any continuous generative network, including normalizing flows and diffusion models. In addition, we show how to assess calibration of mixed posteriors by combining simulation-based calibration (Talts et al., 2018) for continuous parameters with expected calibration error (Guo et al., 2017) for discrete parameters, and propose an empirical finite-sample baseline for discrete calibration. We demonstrate that MNPE performs accurate inference on several problems of increasing difficulty: We first verify that our method works on a tractable mixed Gaussian simulator with an analytical reference solution. We then show that it matches MCMC on a queueing simulator with available likelihoods. Finally, we show that MNPE obtains well-calibrated posteriors for an intractable biophysical neuroscience simulator. The MNPE implementation will be made publicly available upon acceptance.

## 2 Background

### 2.1 Neural Posterior Estimation

Neural Posterior Estimation (NPE) is a powerful method for simulation-based inference (Papamakarios & Murray, 2016; Greenberg et al., 2019). NPE first generates a dataset of parameter–simulation pairs $(\boldsymbol{\theta}, \boldsymbol{x})$ by sampling from the prior $p(\boldsymbol{\theta})$ and running the simulator $\boldsymbol{x} \sim p(\boldsymbol{x} \mid \boldsymbol{\theta})$. It then trains an inference network $q(\boldsymbol{\theta} \mid \boldsymbol{x})$ (i.e., a conditional generative model) to directly approximate the posterior $p(\boldsymbol{\theta} \mid \boldsymbol{x})$. After training, the inference network can be queried at any observation $\boldsymbol{x}_o$ and can directly draw posterior samples, making NPE an amortized method where a single training run supports repeated inference over different observations at negligible cost.

Standard NPE inference networks are designed for continuous parameters only. They parameterize the posterior as a continuous probability density, typically via a normalizing flow or diffusion model defined on

$\mathbb{R}^k$ (Greenberg et al., 2019; Radev et al., 2020; Simons et al., 2023). Such networks are well-suited for smooth posteriors supported on open domains, but are not applicable when the parameter space is discrete or mixed discrete–continuous. That is, when $\boldsymbol{\theta} = (\boldsymbol{\theta}_d, \boldsymbol{\theta}_c)$, with $l$ discrete dimensions $\boldsymbol{\theta}_d \in \mathcal{D} = D_1 \times \cdots \times D_l$, where $D_i$ is a finite set for each discrete dimension $i$, and $k$ continuous dimensions, $\boldsymbol{\theta}_c \in \mathbb{R}^k$.

## 2.2 Autoregressive models

Any multi-dimensional probability distribution can be factorized as the product of its conditionals

$$p(\boldsymbol{z}) = p(z_1, ..., z_M) = p(z_1) \cdot p(z_2|z_1) \cdot \; ... \; \cdot \; p(z_M|z_{1,...,M-1}),$$

with $N$ training samples and $i$ indexes the training sample (omitted from all $z$ to simplify notation). Autoregressive models use this factorization to cast the estimation of a high-dimensional probability distribution $p(\boldsymbol{z})$ into a series of one-dimensional estimation problems. Masked autoregressive density estimators (MADEs) define a feedforward network $q(\boldsymbol{z})$ which take samples as input and return, for every dimension, values that parameterize each conditional probability distribution (i.e., $p(z_1)$, $p(z_2|z_1)$,...). MADEs mask individual weights to enforce the autoregressive property (e.g., $p(z_2|z_1)$ should not depend on $z_{>1}$). With this setup, the joint log-probability can be evaluated in parallel across all dimensions, which enables efficient training with the negative log-likelihood

$$\mathcal{L} = -\frac{1}{N} \sum_i^N \log q(\boldsymbol{z}) = -\frac{1}{N} \sum_i^N \left( \log q(z_1) + \log q(z_2|z_1) + \; ... \; + \; \log q(z_M|z_{1,...,M-1}) \right),$$

where $N$ is the number of training samples and $i$ indexes the training sample. After training, one can draw samples from the MADE by sequentially performing $M$ forward passes (i.e., one forward pass per dimension).

We will use a conditional autoregressive model to estimate the distribution of parameters $\boldsymbol{\theta}$ given simulation outputs $\boldsymbol{x}$. As such, the distribution will be over $\boldsymbol{\theta}$ (instead of $z$), and all distributions will additionally be conditioned on $\boldsymbol{x}$.

## 3 Methods

### 3.1 Inference for mixed parameter spaces

We factorize the posterior over $\boldsymbol{\theta} = (\boldsymbol{\theta}_d, \boldsymbol{\theta}_c)$ as

$$p(\boldsymbol{\theta}_d, \boldsymbol{\theta}_c \mid \boldsymbol{x}) = p(\boldsymbol{\theta}_d \mid \boldsymbol{x})\, p(\boldsymbol{\theta}_c \mid \boldsymbol{\theta}_d, \boldsymbol{x}).$$

This factorization separates the inference problem into two sub-problems: classification over the discrete space $\mathcal{D}$, and conditional density estimation in a purely continuous domain, given the discrete state.

This naturally gives rise to two dedicated inference networks, which together define the *Mixed Neural Posterior Estimation* (MNPE) algorithm. For the discrete factor $p(\boldsymbol{\theta}_d \mid \boldsymbol{x})$, we use an autoregressive network, specifically a Masked Autoregressive Density Estimator (MADE; Germain et al. 2015), which returns the class probabilities for each discrete parameter dimension conditioned on all preceding ones. Therefore, a MADE can handle an arbitrary number of discrete parameters, each with a varying number of classes $|D_i|$. For the continuous factor $p(\boldsymbol{\theta}_c \mid \boldsymbol{\theta}_d, \boldsymbol{x})$, we use a standard conditional generative model (a normalizing flow or diffusion model) conditioned on both $\boldsymbol{\theta}_d$ and $\boldsymbol{x}$. This modular design allows any continuous inference network to be substituted without modifying the discrete component. As with other NPE inference methods, MNPE avoids the need for specialized MCMC kernels or sampler tuning as it targets directly the posterior.

We train our mixed inference network $q(\boldsymbol{\theta}_d, \boldsymbol{\theta}_c \mid \boldsymbol{x})$ using the joint negative log-probability of mixed and discrete parameters $-\log q(\boldsymbol{\theta}_d, \boldsymbol{\theta}_c \mid \boldsymbol{x})$. Using the factorization defined above, the MNPE training objective factorizes as

$$-\log q(\boldsymbol{\theta}_d, \boldsymbol{\theta}_c \mid \boldsymbol{x}) = -\log q(\boldsymbol{\theta}_c \mid \boldsymbol{\theta}_d, \boldsymbol{x}) - \log q(\boldsymbol{\theta}_d \mid \boldsymbol{x}),$$

which separates the loss into a loss over continuous and discrete parameters. For the continuous part, we use a normalizing flow and directly evaluate $\log q(\boldsymbol{\theta}_c \mid \boldsymbol{\theta}_d, \boldsymbol{x})$. For the discrete parameters, the negative

log-likelihood reduces to a cross entropy loss. Finally, we note that the factorization of the loss would also permit other loss functions and, thus, other architectures for the discrete and continuous parameters (e.g., flow-matching or diffusion models for the continuous parameters).

### 3.2 Calibration for mixed posteriors

In the absence of a reference posterior, statistical calibration provides a principled way to assess posterior quality on average: a calibrated posterior requires that events assigned probability $p$ occur with frequency $p$ under the ground truth posterior (Gneiting et al., 2007). Standard calibration tools in SBI, such as simulation-based calibration (SBC) (Cook et al., 2006; Talts et al., 2018) allow checking posterior calibration without requiring direct access to the underlying reference posteriors. However, they assume continuous parameters and do not directly apply to mixed posteriors. We suggest combining rank-based calibration techniques like SBC for the continuous dimensions with established calibration methods from the classification literature for the discrete dimensions.

**Continuous parameters.** SBC generates a calibration set of parameter-data pairs $(\boldsymbol{\theta}_i, \boldsymbol{x}_i) \sim p(\boldsymbol{\theta}) \, p(\boldsymbol{x} \mid \boldsymbol{\theta})$ from the prior and the simulator and obtains approximate posterior samples $\{\boldsymbol{\theta}^{(s)}\}_{s=1}^S \sim q(\boldsymbol{\theta} \mid \boldsymbol{x}_i)$ for each. For a specified scalar projection $f : \boldsymbol{\theta} \to \mathbb{R}$, SBC then checks whether the rank of $f(\boldsymbol{\theta})$ (i.e., the true parameters) is uniformly distributed within $f(\boldsymbol{\theta}^{(s)})$ (i.e., posterior samples). This is a necessary condition for a well-calibrated posterior, i.e., deviation from uniform ranks implies miscalibration in the posterior estimate. A common choice for $f(\cdot)$ is the projection to single dimensions independently, computing a marginal rank statistic per dimension (see also Appendix A.2). Deviations of the distribution over ranks from uniformity are visualized as rank histograms per parameter dimension, or as a cumulative distribution function (CDF) of the empirical rank. We can summarize the deviation from optimality by the error over diagonal (EoD), defined as the mean absolute error of the empirical rank CDF from the diagonal, averaged across dimensions.

**Discrete parameters.** For discrete parameters, rank-based SBC is not applicable because the posterior is a probability mass function rather than a continuous density (Talts et al., 2018). Therefore, we instead use reliability diagrams and the top-label expected calibration error (ECE; Guo et al. 2017): For each discrete dimension $i$ and each test pair $(\boldsymbol{\theta}, \boldsymbol{x})$, we define the predicted class and associated confidence as

$$\hat{\theta}_{d_i} = \arg\max_j q(\theta_{d_i}{=}j \mid \boldsymbol{x}), \qquad \hat{p}_{d_i} = q(\theta_{d_i}{=}\hat{\theta}_{d_i} \mid \boldsymbol{x}),$$

and bin the $N$ test pairs by confidence into $B$ equal-width bins. For a perfectly calibrated posterior, the empirically observed class accuracy matches the confidence of the posterior probability across all bins, i.e., $\mathrm{acc}(b) = \mathrm{conf}(b)$ for every bin $b$, where

$$\mathrm{acc}(b) = \frac{1}{n_b} \sum_{j \in b} \mathbf{1}(\hat{\theta}_{d_i,j} = \theta_{d_i,j}), \qquad \mathrm{conf}(b) = \frac{1}{n_b} \sum_{j \in b} \hat{p}_{d_i,j},$$

with $n_b$ the number of test pairs in bin $b$. The ECE summarizes per-bin deviations as a scalar,

$$\mathrm{ECE}(d_i) = \sum_{b=1}^{B} \frac{n_b}{N} \big| \mathrm{acc}(b) - \mathrm{conf}(b) \big|.$$

Reliability diagrams, typically shown as bar plots of $\mathrm{acc}(b)$ against $\mathrm{conf}(b)$, provide a calibration diagnostic for discrete parameters analogous to SBC rank CDFs for continuous ones: in both cases, agreement with the diagonal indicates calibration.

**Interpreting calibration results.** Together, marginal SBC for continuous $\boldsymbol{\theta}_c$ and marginal ECE for discrete $\boldsymbol{\theta}_d$ provide complementary calibration checks for all dimensions of a mixed posterior. These diagnostics can be interpreted both visually and quantitatively. A well-calibrated posterior yields uniform rank CDFs and diagonal reliability diagrams, with EoD and ECE close to zero. Visually, specific shapes in the SBC

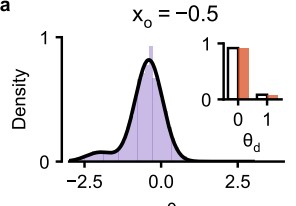
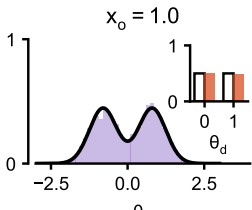
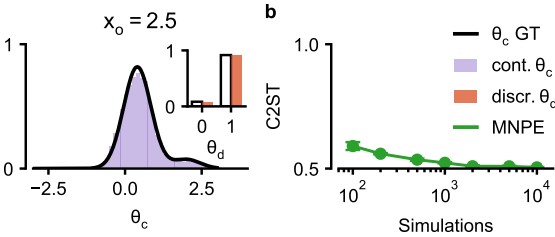

Figure 2: **Tractable Gaussian benchmark.** **(a)** MNPE posteriors for three observations spanning different regimes: $x_o = -0.5$ (discrete state $\boldsymbol{\theta}_d$=0 dominates, unimodal), $x_o = 1.0$ (uncertain $\boldsymbol{\theta}_d$, bimodal), and $x_o = 2.5$ ($\boldsymbol{\theta}_d$=1 dominates). Histograms show the continuous marginal posterior $p(\theta_c \mid x_o)$ from MNPE samples (violet), with the analytical ground-truth density overlaid (black). Inset bar charts compare the discrete posterior $P(\boldsymbol{\theta}_d \mid x_o)$ between GT (black outline) and MNPE (orange). **(b)** C2ST on the joint $(\theta_c, \boldsymbol{\theta}_d)$ distribution as a function of training simulations. Error bars show standard deviations over five independent training runs, each evaluated on ten held-out test observations. The score approaches the chance level of 0.5 with ∼1,000 simulations.

rank histograms reveal miscalibration patterns, e.g., u-shaped histograms or above diagonal CDFs indicate overconfidence (Talts et al., 2018); for reliability diagrams, bars below the diagonal (confidence exceeds accuracy) indicate overconfidence, while bars above indicate underconfidence.

Quantitatively, interpreting EoD and ECE values requires finite-sample baselines, since both metrics remain strictly positive even under perfect calibration due to finite sample noise. For the EoD, the expected deviation under uniform ranks is known analytically, independent of the posterior estimator, and provides a baseline with an associated confidence band (Talts et al., 2018). For the ECE, no such posterior-independent baseline exists: the finite-sample noise floor depends on how test cases are distributed across confidence bins, which is determined by the posterior estimator itself and therefore varies with training budget. Therefore, to distinguish actual miscalibration in discrete posterior dimensions from statistical noise, we propose an empirical baseline under the assumption of a perfectly calibrated classifier (Appendix A.1).

## 4 Results

We validate our approach on three simulators with increasing complexity. First, we showcase MNPE on a mixed Gaussian simulator which is analytically tractable and allows for an analytical reference solution. Second, we apply MNPE to a queueing simulator with known likelihood, and compare it to an MCMC reference solution (additional comparison to a common mixed MCMC task in Appendix B.1). In the last example we turn to a 'black-box' simulator with intractable likelihood, the Hodgkin–Huxley simulator.

We use three complementary metrics to measure the quality of the approximated posteriors. When reference posteriors are available, we use the *Classifier two-sample test* (C2ST; Friedman 2004; Lopez-Paz & Oquab 2017; Lueckmann et al. 2021), which trains a classifier to distinguish MNPE posterior samples from a reference distribution. A score near 0.5 (chance level) indicates the two distributions are indistinguishable. To test statistical calibration in the absence of a reference, we use SBC for continuous posterior dimensions and ECE for discrete parameters.

### 4.1 Gaussian example

We construct a tractable toy example with one continuous and one discrete latent variable coupled through a shared Gaussian observation

$$\boldsymbol{\theta}_c \sim \mathcal{N}(0, 1),$$
$$\boldsymbol{\theta}_d \sim \text{Bernoulli}(0.5),$$
$$x \mid \boldsymbol{\theta}_c, \boldsymbol{\theta}_d \sim \mathcal{N}(\boldsymbol{\theta}_c + a\,\boldsymbol{\theta}_d, \ \sigma^2),$$

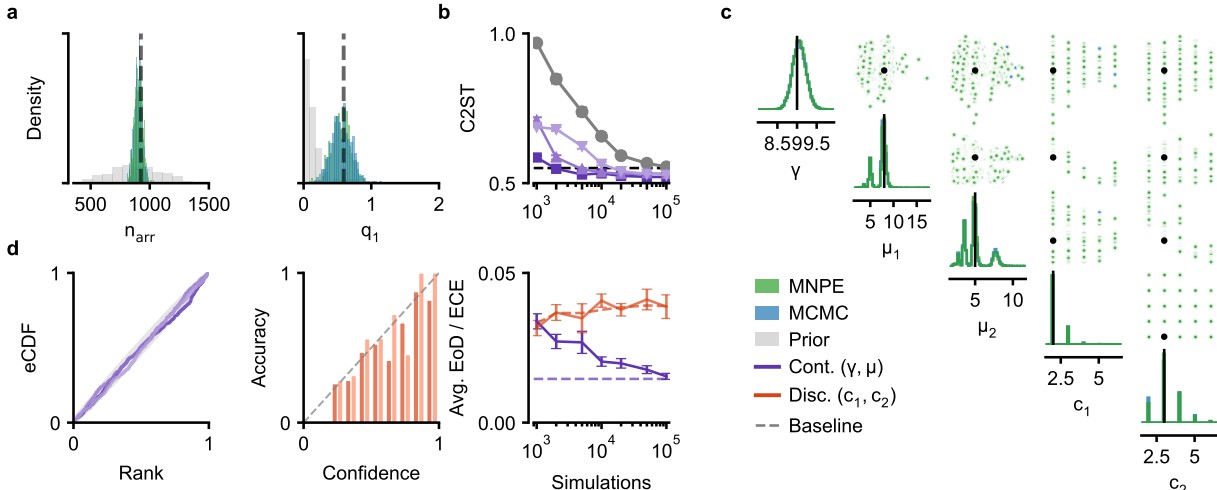

Figure 3: **Tandem queueing simulator**. **(a)** Posterior predictive simulations for arrival counts $n_{\mathrm{arr}}$ and queue length $q_1$, comparing MNPE (green), MCMC (blue), and prior predictive (grey). Dashed line marks $x_o$. **(b)** C2ST for joint (grey) and marginal comparisons (violet) against the MCMC reference across training samples. Error bars show standard error of the mean over five MNPE training repetitions. Dashed black line: inter-reference baseline (PyMC vs. NumPyro). **(c)** Joint posterior: MNPE (green) versus MCMC reference (blue). Black markers indicate ground-truth values. **(d)** Calibration diagnostics at $10^5$ simulations. *Left:* SBC rank eCDF for the continuous parameters ($\gamma$, $\mu_1$, $\mu_2$), with 95% uniform confidence band in grey. *Center:* ECE reliability diagram for the discrete parameters ($c_1$, $c_2$). *Right:* Expected calibration error for continuous (EoD, violet) and discrete (ECE, orange) parameters, averaged across five seeds and all marginals across budgets. Dashed lines indicate expected optimal calibration error for the given training budget.

with shift $a = 2$ and noise $\sigma = 0.5$. The discrete parameter $\boldsymbol{\theta}_d$ shifts the observation mean, making the marginal posterior $p(\boldsymbol{\theta}_c \mid x)$ a two-component Gaussian mixture whose weights depend on $x$. Specifically, $p(\boldsymbol{\theta}_d = 1 \mid x)$ follows a logistic form and the conditional $p(\boldsymbol{\theta}_c \mid \boldsymbol{\theta}_d, x)$ is Gaussian with closed-form mean and variance, yielding an analytical reference posterior. This example can be interpreted as a noisy channel simulator, where the receiver must jointly infer the transmitted signal $\boldsymbol{\theta}_c$ and the channel mode $\boldsymbol{\theta}_d$ from a single noisy observation. This tractable toy example captures the basic challenge of mixed-parameter inference: discrete-continuous coupling that induces multimodal continuous posteriors.

We train MNPE on increasing simulation budgets $N = 100, \ldots 10{,}000$ using a neural spline flow for the continuous parameters and a categorical MADE for the discrete parameters. We compute MNPE posteriors for three test observations spanning different regimes of the discrete variable (Figure 2a): $\boldsymbol{\theta}_d = 0$ dominates for small $x$, $\boldsymbol{\theta}_d = 1$ dominates for large $x$, and the posterior is bimodal at intermediate values. MNPE accurately recovers the ground-truth posterior in all cases, including the bimodal regime where both discrete states are plausible ($x_o = 1.0$). The inset bar charts confirm that the discrete marginal $P(\boldsymbol{\theta}_d \mid x_o)$ is well calibrated across regimes (Figure 2a). We additionally compute the C2ST score to the ground truth posterior, based on 1,000 posterior samples. The C2ST score converges to the chance level of 0.5 at around $\sim$1,000 training simulations, indicating that MNPE provides accurate posterior estimates (Figure 2b).

## 4.2 Tandem queueing simulator

We apply MNPE to a tandem queueing simulator from operations research Jackson (1957), consisting of two M/M/$c$ queues in series (Gross et al., 2008) where customers arrive at station 1 and, after being served, proceed to station 2. The inference problem has five parameters: three continuous arrival and service rates $\boldsymbol{\theta}_c = (\gamma, \mu_1, \mu_2)$ and two discrete server counts $\boldsymbol{\theta}_d = (c_1, c_2) \in \{2, \ldots, 6\}^2$ ($|\mathcal{D}| = 25$). The simulator produces

a five-dimensional observation comprising arrival counts, completion counts, and queue lengths over a fixed time horizon (see Appendix C for details).

For this simulator, no analytical posterior is available, but the closed-form likelihood enables MCMC-based inference, which we use as a reference solution. Since gradient-based samplers such as NUTS (Hoffman & Gelman, 2014) cannot directly handle discrete parameters, we marginalize out the discrete dimensions analytically, reducing inference to a continuous sampling problem. Because MCMC inference in this setting is challenging, we construct two independent references with different marginalization procedures: We use PyMC (Abril-Pla et al., 2023) with automatic marginalization and NumPyro (Phan et al., 2019) with manual marginalization, and cross-validate them against each other to confirm correctness (details in Appendix C). Both MCMC references agree closely (C2ST $\approx$ 0.55), providing confidence that both recover the correct posterior.

We select the MNPE inference network architecture via hyperparameter search using Optuna (Akiba et al., 2019), optimizing the negative log-probability of the current posterior estimate on a held-out validation set (Lueckmann et al., 2021). The resulting network consists of a categorical MADE for the discrete parameters and a neural spline flow (NSF) for the continuous parameters (architecture details in Appendix C). We train MNPE with up to $N = 100{,}000$ simulations.

In the posterior predictive space, the MNPE-simulated observations match the MCMC reference and concentrate around the true observation for a training budget of $N = 100{,}000$ simulations (Fig. 3a). For low simulation budgets, we observe that MNPE posteriors deviates substantially from the MCMC reference in terms of C2ST score (Fig. 3b), but the C2ST decreases monotonically with increasing simulation budget and converges to the C2ST score we observed between the two MCMC references. For a single example observation $\boldsymbol{x}_o$ of arrival counts and queue lengths, MNPE recovers a posterior that closely matches the MCMC reference across both continuous and discrete parameters and identifies the true server configuration ($c_1 = 2, c_2 = 3$) as the dominant mode (Fig. 3c). The MNPE posterior based on $N = 100{,}000$ training simulations is well-calibrated: rank statistics for the continuous parameters are approximately uniform (Fig. 3d, left), and reliability diagrams confirm that predicted class probabilities closely track empirical accuracy for the discrete parameters (Fig. 3d, center). Across increasing MNPE training budgets, calibration improves consistently for the continuous dimensions, while it stays stable for the discrete dimensions, closely matching the expected optimal calibration (Fig. 3d, right).

### 4.3 Hodgkin–Huxley simulator

As a final example, we apply MNPE to the Hodgkin–Huxley simulator (Hodgkin & Huxley, 1952), which describes neural activity through the dynamics of ion channels. Different channels can produce qualitatively distinct behaviours, such as spike-frequency adaptation in the presence of $M$-type potassium channels ($K_m$) or diverse bursting patterns induced by calcium channels (Pospischil et al., 2008). Typically, inference focuses on the maximal conductances $g_i$ of the channels, and channel-identities are assumed to be known. However, in practice, the exact combination of channels in a neuron is often unknown. Here, we apply MNPE to simultaneously infer continuous channel densities and categorical channel identities. While a large variety of ion channels exists (Podlaski et al., 2017), we focused on a minimal set consisting of $Na$ and $K$ channels, optional $K_m$, and $L$- and $T$-type calcium channels ($Ca_L$, $Ca_T$), which together yield rich dynamical regimes (Pospischil et al., 2008).

We implemented the simulator using the JAXLEY simulator (Deistler et al., 2025b) and added observational noise to the simulated voltage trace (Fig. 4a). To reduce the dimensionality of the time-series we summarized it by a set of 14 features, such as first and second moments computed over different time intervals, mirroring previous work (Gonçalves et al., 2020) (details in Appendix C). This results in an inference problem with four continuous parameters and two discrete variables: The continuous variables modulate the sodium, potassium, and leak conductances ($g_{Na}, g_K, g_L$), as well as the leak reversal potential ($E_L$). The two discrete parameters indicate the presence of slow potassium dynamics $K_m$ and select the calcium channel type ($Ca_L$, $Ca_T$, or none) with fixed parametrization.

We train MNPE on 100,000 simulations and perform inference given a synthetically generated observation. We found that samples from the posterior closely match the observation qualitatively, with similar spiking

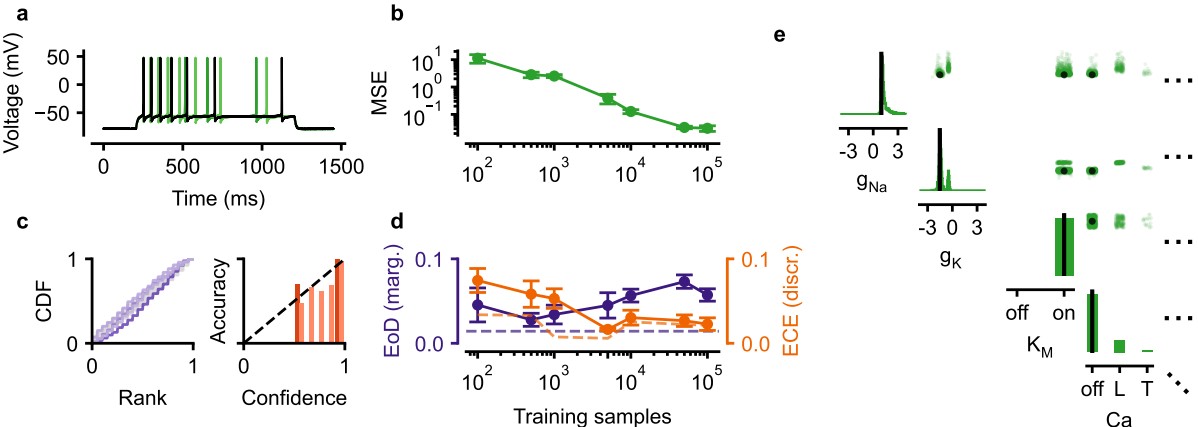

Figure 4: **MNPE on a Hodgkin-Huxley simulator.** **(a)** One observation (black) and two posterior predictive voltage traces (green) for a posterior trained on 100k samples. **(b)** MSE (mode ±std.) on 1k test samples. **(c)** Simulation-based calibration for a posterior trained on 100k samples, *Left:* for continuous parameters, *Right:* for two discrete parameters. **(d)** Simulation-based calibration across training samples. Mean error of diagonal (EoD) on the marginal distributions for the continuous parameter as well as mean expected calibration error (ECE) for the discrete parameter dimensions. Error bars show std. across three random seeds. **(e)** One and two dimensional marginal posterior distribution for four selected parameters for the observation shown in a. Black dots/lines indicate ground truth parameters. Continuous parameters are presented on a normalized scale (see Fig. B-2 for all dimensions).

characteristics (Fig. 4a). We then evaluate the performance of MNPE with respect to the amount of available training data. As expected, the mean squared error (MSE) of the summary statistics decreases with increasing numbers of training samples (Fig. 4b).

Next, we evaluate the calibration of the MNPE posterior. While the expected calibration error is slightly higher in the low-data regime, the calibration of the continuous parameters remains stable across training set sizes, indicating good calibration across all parameter dimensions (Fig. 4c,d). Interestingly, the discrete parameter for $K_m$ mainly occupies two bins in the ECE diagram, which indicates that the posterior has either high or low confidence, but is well calibrated in both cases (Fig. 4c, left). Having a joint posterior over continuous and discrete parameters in this situation allows to draw conclusions about the presence of specific ion channels and additionally enables the analysis of compensatory mechanisms and interactions between ion channels. For example, for this particular simulation of a low-frequency spiking neuron with temporal adaptation, we observe that the presence of an *L*-type calcium channel may be associated with higher potassium conductance $g_K$ (Fig. 4e).

## 5 Discussion

We presented MNPE, an extension of neural posterior estimation to simulators with mixed discrete and continuous parameters. Across three examples of increasing complexity, the approach produces posterior estimates that match MCMC references where available and remain well-calibrated for both continuous and discrete parameter dimensions. To assess calibration in the mixed setting, we combined rank-based diagnostics for continuous parameters with classification-based metrics for discrete parameters and derived finite-sample baselines that allow reliable interpretation of both.

**Related work.** Our work is inspired by previous work on performing inference in simulators with mixed simulation *outputs*, namely mixed likelihood estimation (MNLE, Boelts et al., 2022). While MNLE factorizes the observations (i.e., *likelihoods*) into discrete and continuous components, MNPE targets applications with mixed *parameter spaces*, and directly the posterior. In addition, our work extends the inference network to

multiple categorical variables by using categorical MADEs. In a similar spirit to our discrete–continuous factorization, dedicated methods have been proposed for hierarchical models. Rodrigues et al. (2021) introduce an NPE method for hierarchical model structures in which observations share certain global parameters, while Habermann et al. (2025) extend this to general multilevel hierarchical models. Although both approaches share the core idea of factorization with separate networks for global and local parameters, they are restricted to continuous parameters.

Several recent methods address the mixed inference problem using diffusion models. Ghiglino et al. (2026) adopt the same conditional factorization as MNPE but model both components with diffusion processes, requiring a Riemannian continuous embedding for the discrete space. Schröder & Macke (2024) treat $\boldsymbol{\theta}_d$ as a model index and use simple mixture-based posteriors to perform inference over both, alternative model components and their associated parameters. Gloeckler et al. (2026) extend joint inference setting to a transformer-based encoder-decoder architecture combined with a diffusion process over the continuous parameters, resulting in a more flexible but substantially heavier framework. Compared to these approaches, MNPE relies on a masked autoregressive estimator and a normalizing flow, which makes training simpler and enables posterior evaluation and sampling in a single forward pass rather than through iterative denoising. However, for high-dimensional posteriors, large simulation budgets, or scenarios where inference time is less critical, diffusion-based methods may still be advantageous. In the direction foundation models, mixed discrete–continuous tabular diffusion models (Shi et al., 2025) and in-context learning approaches (Vetter et al., 2025) show promise for heterogeneous feature types and could motivate future alternatives.

While MCMC methods are a natural baseline for posterior inference in the setting of available likelihoods, handling mixed discrete and continuous parameters remains challenging. Discontinuous HMC (Nishimura et al., 2020) embeds discrete parameters into a continuous space and simulates Hamiltonian dynamics on a piecewise-smooth density, but is limited to ordinal discrete parameters and does not generalize to arbitrary discrete state spaces. Mixed HMC (Zhou, 2020) evolves discrete and continuous variables jointly within HMC trajectories, yet still requires a differentiable likelihood and offers no advantage over analytically marginalizing out the discrete dimensions. Furthermore, marginalization cost scales as $O(|\mathcal{D}|)$ per MCMC step, which becomes increasingly costly for larger discrete spaces, a limitation that MNPE does not share. Both MCMC approaches furthermore require tractable likelihood and gradient evaluations and do not amortize—every new observation demands a full sampling run. MNPE, in comparison, can perform inference for any black-box simulator and directly draws samples from the posterior distribution in a single forward pass, thereby amortizing the cost of inference.

**Limitations.** As our work is a method in the NPE family, it also inherits its weaknesses: it may need a large training dataset to provide accurate estimations for complex posterior structures (e.g. (Deistler et al., 2022b)), it is susceptible to model misspecification (Cannon et al., 2022; Kelly et al., 2025), and while it targets the posterior across the full prior range, one might only be interested in a specific observation. In such a case, multi-round NPE approaches (Greenberg et al., 2019) can be adopted. A limitation in the presented calibration procedure is that we only assess marginal calibration for each dimension separately. While we could use the "joint marginals" $p(\boldsymbol{\theta}_c \mid \boldsymbol{\theta}_d, \boldsymbol{x})$ on the continuous dimensions, the full mixed posterior is not appropriate for standard expected coverage calibration checks (Deistler et al., 2022a). Furthermore, the empirical ECE baseline relies on the half-normal approximation to the binomial, which can underestimate the noise floor at very small test set sizes (Appendix A.1).

**Conclusion.** Simulators with discrete parameters are prevalent throughout the sciences and engineering; yet their specifics are often overlooked when designing inference and calibration methods for simulation-based inference. We presented a strategy to build inference networks addresses this. We showed the effectiveness of this method across multiple examples and we will provide an open source implementation upon acceptance.

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

# Appendix Contents

## A Calibration checks

### A.1 Empirical baselines for discrete calibration checks

To quantify the noise floor for the ECE, we consider $B$ equal-width confidence bins and let $n_b$ denote the number of test cases in bin $b$ with center $p_b$ (approximating the mean confidence in the bin). Under perfect calibration, each test case in bin $b$ is classified correctly with probability $p_b$. The count of correct predictions therefore follows a binomial distribution $n_b \cdot \mathrm{acc}(b) \sim \mathrm{Binomial}(n_b, p_b)$ and the per-bin contribution to the expected calibration error is

$$\frac{n_b}{N} \mathbb{E}\big[|\,\mathrm{acc}(b) - p_b|\big] = \frac{n_b}{N} \sum_{k=0}^{n_b} |\frac{k}{n_b} - p_b| \binom{n_b}{k} p_b^k (1-p_b)^{n_b-k} \ .$$

Since the calculation of this sum can get expensive for large $n_b$, we can approximate it via a half-normal distribution. A commonly used rule of thumb is $n_b p_b \geq 5$ and $n_b(1-p_b) \geq 5$ for a reliable approximation. Applying this approximations yields

$$\mathbb{E}\big[|\,\mathrm{acc}(b) - p_b|\big] \approx \sqrt{\frac{2}{\pi}} \sqrt{\frac{p_b(1-p_b)}{n_b}} \,,$$

and the expected ECE under perfect calibration reduces to

$$\mathbb{E}[\mathrm{ECE}] = \frac{1}{N} \sqrt{\frac{2}{\pi}} \sum_b \sqrt{n_b \cdot p_b(1-p_b)}\,.$$

This formula depends on the bin occupancies $\{n_b\}$, which is the occupancy under the generally unknown true posterior. A naive ansatz could use a uniform occupancy, which tends to produce a higher baseline because the high binomial variance-bins in the center weighted equally in all cases. Therefore, we suggest using a tighter baseline that uses empirical bin counts obtained from the approximate posterior estimate (Widmann et al., 2019; Vaicenavicius et al., 2019). Importantly, the posterior estimator is used only for obtaining bin occupancy, not for the confidences themselves ($p_b$ is defined by the center of each bin). Note that because the bin counts depend on the posterior estimator, this baseline varies with varying confidence of the posterior estimator, e.g., with varying training budgets. To avoid overly noisy estimates in bins with low occupancy, previous work replaced the equal-width bins with equal-mass (adaptive quantile) bins (Nixon et al., 2019). However, we found that the equal-mass binning approach produced systematically higher baselines and therefore used the empirical baseline based on equal-width bins and posterior predicted bin occupancies throughout our experiments.

It is worth noting that both empirical baselines for ECE and EoD depend on the number of calibration samples $N$, which should match the number used to evaluate the calibration of the posterior. Since computing calibration requires $N$ simulations $\boldsymbol{x} \sim p(\boldsymbol{x} \mid \boldsymbol{\theta})$, this number is typically limited to a few hundred (we use $N = 500$ in all experiments). At the same time, increasing $N$ yields tighter baselines; in the limit $N \to \infty$, the baseline converges to zero, implying that any approximate posterior will exhibit some deviation from perfect calibration.

### A.2 Alternative Calibration checks

Instead of investigating the calibration of each marginal distribution individually, other functions can be used that project into one dimension (Talts et al., 2018). The canonical projection using the posterior log-probability (Deistler et al., 2022a), however, fails in the mixed-parameter setting. The theoretical guarantee that rank statistics are uniform under a well-calibrated posterior relies on the posterior being absolute continuous (Talts et al., 2018). This condition is violated by discrete components. In practice, the discrete components tend to dominate the joint log-probability, masking miscalibration in the continuous parameters. An alternative approach is to separate the posterior into its discrete and continuous components. For the continuous parameters we can then use the log-probability on the "joint marginal" of all continuous

dimensions $p(\boldsymbol{\theta}_c \mid \boldsymbol{x})$ as projection function while still applying marginal calibration to the discrete dimensions as before. However, to be consistent across dimensions $i, j$, we assess calibration of the marginals of $\boldsymbol{\theta}_c^i$ and $\boldsymbol{\theta}_d^j$ separately as described in Section 3.2.

## B    Additional examples

### B.1    Coal mining disaster changepoint inference

The coal mining switchpoint simulator is presented in the PyMC documentation[1] as an example for dealing with discrete model parameters using automatic marginalization. We include it here as a second direct comparison between amortized MNPE and MCMC inference. This example differs from the queueing model in two key respects: the observation is high-dimensional ($x \in \mathbb{N}^{111}$, one count per year), requiring an embedding network, and the discrete space is substantially larger ($|\mathcal{D}| = 111$ switchpoints versus $|\mathcal{D}| = 25$ server configurations). At the same time, this inference problem is arguably easier: the switchpoint only determines where the rate changes, inducing weaker discrete–continuous coupling than the queueing simulator's stability constraint, and the posterior is unimodal.

The coal mining disaster dataset records the number of coal mining disasters per year in the United Kingdom from 1851 to 1961 (Jarrett, 1979). A visible decline in disaster frequency around 1890, attributed to improved safety regulations, makes this a classic benchmark for Bayesian changepoint analysis (Raftery & Akman, 1986). While a continuous relaxation of the switchpoint is possible for this model, we keep the discrete formulation as a representative test case for the broader class of models with inherently discrete parameters.

The observed disaster count $y_t$ in year $t$ follows a Poisson likelihood with a rate that switches at an unknown year $s$:

$$s \sim \text{DiscreteUniform}(1851, 1961),$$
$$\lambda_{\text{early}}, \lambda_{\text{late}} \sim \text{Exponential}(1),$$
$$y_t \mid s, \lambda_{\text{early}}, \lambda_{\text{late}} \sim \text{Poisson}\big(\lambda_{\text{early}} [t < s] + \lambda_{\text{late}} [t \geq s]\big).$$

In the notation of Section 3, the discrete parameter is $\theta_d = s \in \{1851, \ldots, 1961\}$ ($|\mathcal{D}| = 111$) and the continuous parameters are $\theta_c = (\lambda_{\text{early}}, \lambda_{\text{late}}) \in \mathbb{R}_{>0}^2$. The observation is the full sequence of annual disaster counts $x = (y_{1851}, \ldots, y_{1961}) \in \mathbb{N}^{111}$.

The discrete switchpoint is analytically marginalized reducing inference to a purely continuous problem amenable to NUTS (Abril-Pla et al., 2023; Hoffman & Gelman, 2014). The discrete posterior $p(s \mid x)$ is then recovered by Rao-Blackwellization: for each posterior draw of $(\lambda_{\text{early}}, \lambda_{\text{late}})$, the conditional $p(s \mid \lambda_{\text{early}}, \lambda_{\text{late}}, x)$ is computed analytically via the Poisson log-likelihood summed over all years. The marginalization cost scales as $O(|\mathcal{D}|)$ per MCMC step, requiring 111 Poisson log-likelihood evaluations per gradient computation. While tractable for this simulator, this cost becomes prohibitive for larger discrete spaces.

We performed a hyperparameter search to select an optimal MNPE architecture via Optuna (Akiba et al., 2019), optimizing the negative log-probability of the current posterior estimate (NLTP) on a held-out validation set following Lueckmann et al. (2021). The tuned architecture uses a neural spline flow with 2 coupling transforms, 1 hidden layer of 64 units, and 10 rational-quadratic spline bins. Since the observation $x \in \mathbb{N}^{111}$ is high-dimensional relative to the three-dimensional parameter space, a fully-connected embedding network (1 hidden layer of 64 units, output dimension 32) compresses the observation before the density estimator. We also applied a variance-stabilizing $\sqrt{\cdot}$ transform to the Poisson counts before training, which produces a more uniform variance across rate regimes and improves z-scoring during training. We evaluate MNPE across seven training budgets from $10^3$ to $10^5$ simulations, with five independent training runs per budget.

The cumulative disaster count exhibits a clear change in slope around 1890, consistent with the historical switchpoint. Posterior predictive cumulative curves from MNPE and MCMC both tightly envelop the observed trajectory, whereas the prior predictive band spans a much wider range (Fig. B-1a). With increasing

---

[1] https://www.pymc.io/projects/examples/en/latest/howto/marginalizing-models.html

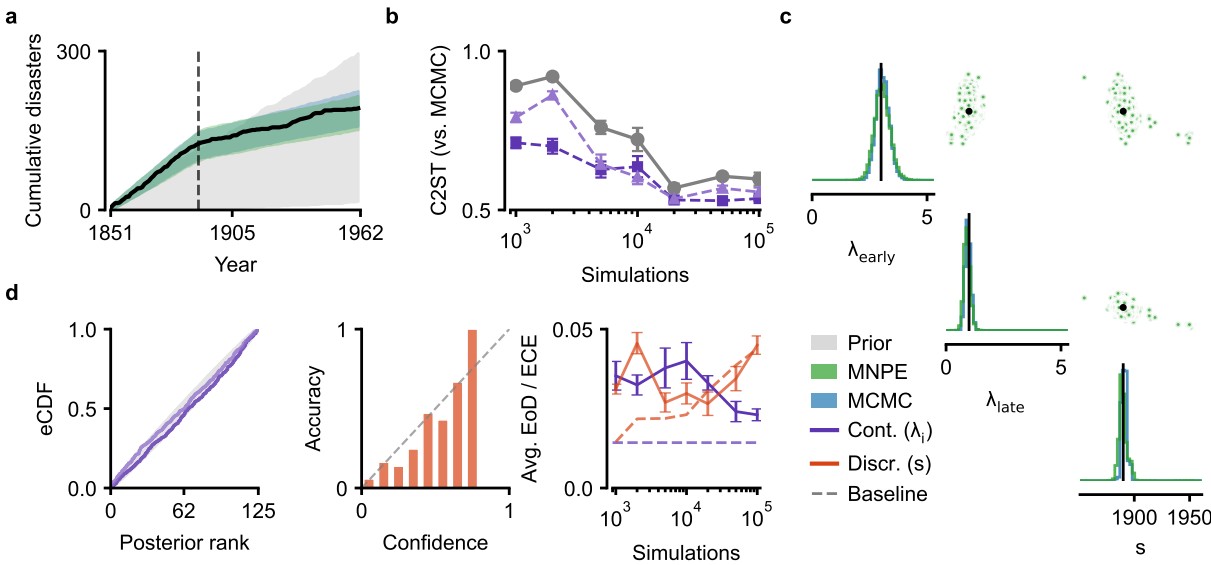

Figure B-1: **Coal mining changepoint simulator** ($|\mathcal{D}| = 111$). **(a)** Cumulative coal mining disasters in the UK (1851–1962, black). The change in slope around 1890 (vertical line) reflects improved safety regulations. Predictive distributions (5–95th percentile) are shown for prior (grey), MNPE (green) and MCMC reference (blue). **(b)** C2ST versus training budget for joint and per-parameter comparisons against the MCMC reference. Error bars show $\pm 1$ SEM over five independent training runs. **(c)** Joint posterior over all three parameters: MNPE (green) versus marginalized MCMC (blue). Black markers indicate historical estimates. **(d)** Calibration diagnostics. *Left:* SBC rank eCDF for the continuous parameters ($\lambda_{\text{early}}$, $\lambda_{\text{late}}$) at $10^5$ simulations, with 95% uniform confidence band in grey. *Center:* ECE reliability diagram for the discrete switchpoint at $10^5$ simulations. *Right:* Expected calibration error for continuous (EoD, violet) and discrete (ECE, orange) parameters, averaged across five seeds and all marginals across budgets. Dashed lines indicate expected optimal calibration error for the given training budget.

training budget, the gap between MNPE and MCMC narrows: joint C2ST falls to $0.56 \pm 0.01$ at $10^5$ simulations, and the marginal C2ST for both rate parameters approaches the chance level of 0.5 (Fig. B-1b). Examining a single observation $\boldsymbol{x}_o$ confirms these results: the MNPE posterior concentrates $\lambda_{\text{early}}$ around 3.0 disasters per year and $\lambda_{\text{late}}$ around 0.9, with the switchpoint peaking near 1890, all closely matching the marginalized MCMC reference (Fig. B-1c). SBC rank distributions at $10^5$ simulations show no systematic deviation from uniformity for either continuous parameter (Fig. B-1d, left), and the ECE reliability diagram for the 111-class switchpoint indicates well-calibrated discrete predictions (Fig. B-1d, center). Across training budgets, continuous calibration steadily improves toward the finite-sample baseline, while discrete calibration remains close to the expected calibration of an perfectly calibrated classifier (Fig. B-1d, right).

## B.2 Hodgkin–Huxley simulator

Below we show the full posterior distributions from Fig. 4.

# C Experimental details

## C.1 Gaussian example

MNPE uses a neural spline flow with 5 coupling transforms and 4 blocks for the continuous parameters, and a MADE for the discrete parameters. Both networks have 3 hidden layers with 32 hidden features each. Training uses up to $N = 10,000$ simulations with learning rate set to $\eta = 5e^{-4}$, validation set fraction of ten percent, and early stopping after 20 epochs without validation loss improvement.

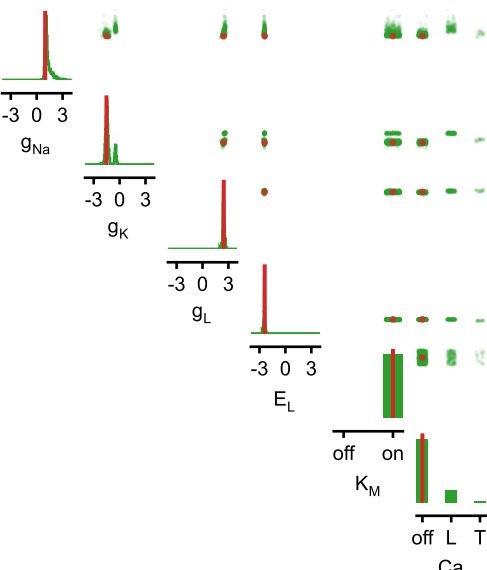

Figure B-2: **Hodgkin-Huxley posterior.** One and two dimensional posterior marginals of the full parameter space (four continuous and two discrete dimensions) for the observation shown in Fig. 4a.

## C.2 Tandem queueing simulator

The tandem queueing simulator consists of two M/M/$c$ queues in series (Gross et al., 2008; Jackson, 1957), where M/M/$c$ denotes a queue with memoryless (Poisson) arrivals, memoryless (exponential) service times, and $c$ parallel servers. Customers arrive at station 1 with rate $\gamma$, are served by $c_1$ parallel servers at rate $\mu_1$, and then proceed to station 2 with $c_2$ servers at rate $\mu_2$. The continuous parameters are the arrival and service rates $\boldsymbol{\theta}_c = (\gamma, \mu_1, \mu_2) \in \mathbb{R}^3_{>0}$ and the discrete parameters are the server counts $\boldsymbol{\theta}_d = (c_1, c_2) \in \{2, 3, 4, 5, 6\}^2$ ($|\mathcal{D}| = 25$). The parameters are drawn from the following prior:

$$\gamma \sim \mathrm{LogNormal}(\log 9, \ 0.3),$$
$$\mu_1 \sim \mathrm{LogNormal}(\log 8, \ 0.3),$$
$$\mu_2 \sim \mathrm{LogNormal}(\log 5, \ 0.3),$$
$$c_1, c_2 \sim \mathrm{DiscreteUniform}\{2, 3, 4, 5, 6\}.$$

The per-station traffic intensity is $\rho_i = \gamma/(c_i\mu_i)$ and both stations must satisfy $\rho_i < 1$ for the system to be stable. In steady state, the expected number of customers waiting in the queue at station $i$ is given by the standard M/M/$c$ result (Gross et al., 2008):

$$\mathbb{E}[Q_i] \;=\; \frac{r_i^{c_i}\, \rho_i}{c_i!\,(1-\rho_i)^2}\, \pi_{0,i}\,, \qquad \text{where} \quad \pi_{0,i} = \left[\sum_{n=0}^{c_i-1} \frac{r_i^{\,n}}{n!} \;+\; \frac{r_i^{c_i}}{c_i!\,(1-\rho_i)}\right]^{-1}$$

with offered load $r_i = \gamma/\mu_i$. By Jackson's theorem, the two stations are independent in steady state, so the joint observation likelihood factorizes across stations. This simulator exhibits strong discrete–continuous coupling: expected queue lengths grow steeply as $\rho_i \to 1$, making the posterior sensitive to the interaction between server counts and service rates.

Given a time horizon $T = 100$, the simulator produces five observations: arrival counts, completion counts at each station, and two queue lengths. The three count dimensions follow Poisson distributions with rate $\gamma T$:

$$n_{\mathrm{arr}} \sim \mathrm{Poisson}(\gamma T), \qquad n_{\mathrm{comp},1} \sim \mathrm{Poisson}(\gamma T), \qquad n_{\mathrm{comp},2} \sim \mathrm{Poisson}(\gamma T).$$

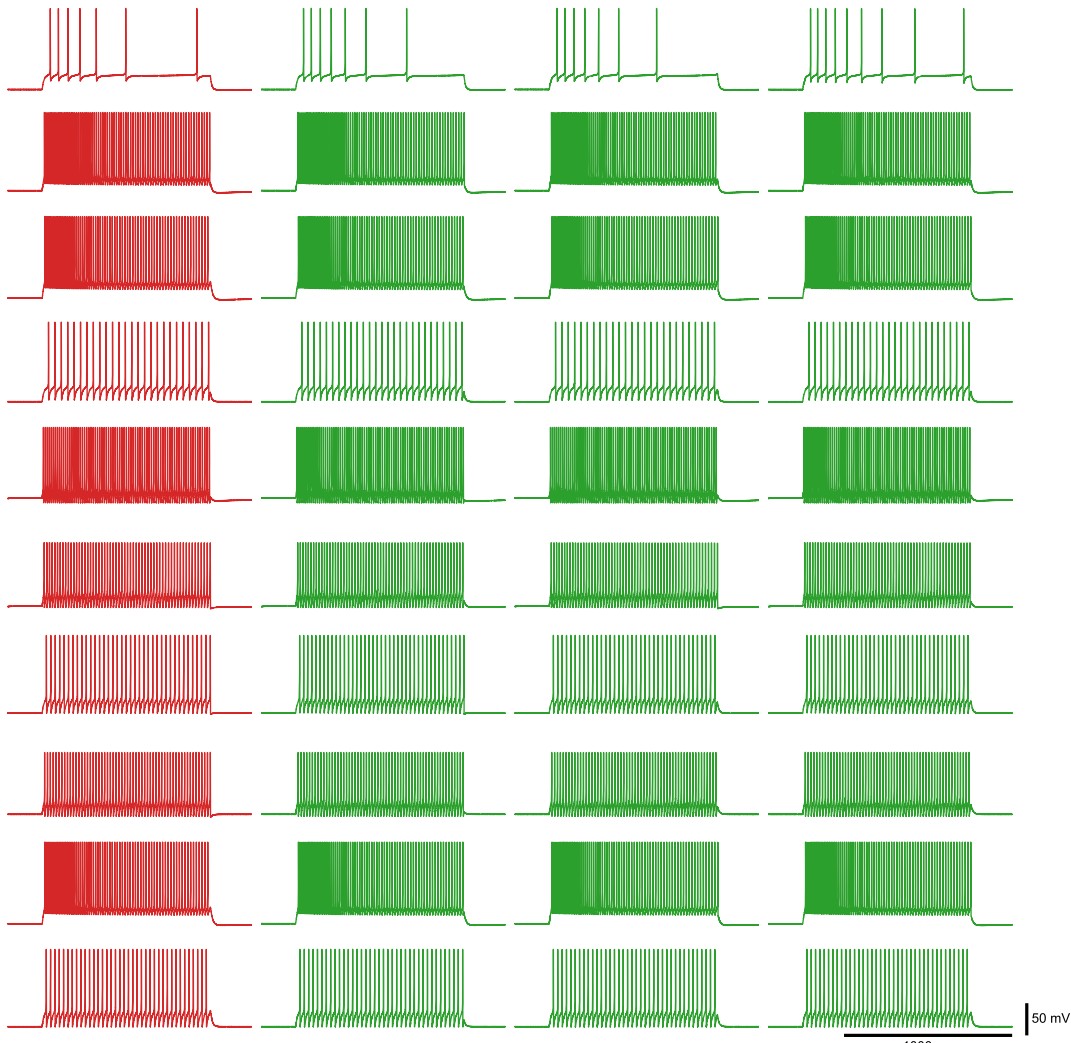

Figure B-3: **Posterior predictives for the Hodgkin-Huxley simulator.** Three posterior predictive samples (green) for 10 observations (red).

In steady state, the throughput at each station equals the arrival rate, so all three counts share the same rate and identify only $\gamma$. The two queue length observations provide the information needed to identify the service rates $\mu_1$ and $\mu_2$, since $\mathbb{E}[Q_i]$ depends nonlinearly on $\rho_i = \gamma/(c_i\mu_i)$:

$$q_i \sim \text{TruncatedNormal}\big(\mathbb{E}[Q_i], \ \sigma_{\text{obs}}, \ \text{lower=0}\big), \qquad \sigma_{\text{obs}} = 0.1.$$

Prior draws that violate the stability condition ($\rho_i \geq 1$) or produce near-unstable systems ($\mathbb{E}[Q_i] > 10$) are discarded during training data generation, removing approximately 4% of samples.

Since NUTS cannot directly handle discrete parameters, both MCMC references marginalize out the discrete dimensions analytically. We implement two independent procedures:

- *PyMC (Abril-Pla et al., 2023).* The 25 discrete configurations are automatically summed analytically to produce a continuous-only mixture log-likelihood which is then sampled with NUTS. Rao-Blackwellization Blackwell (1947) subsequently recovers the discrete posterior from the continuous chains. All this is handled automatically by PyMC routines.

- *NumPyro (Phan et al., 2019).* We run separate NUTS chains for each of the 25 discrete configurations, sampling the unimodal conditional posterior $p(\boldsymbol{\theta}_c \mid c_1, c_2, x)$. The per-configuration results are manually combined via Rao-Blackwell pooling, weighting each chain by its estimated marginal likelihood.

Both approaches require careful numerical treatment to avoid divergences near the stability boundary where $\rho \to 1$. We use dense mass matrices, stability guards, and a high target acceptance rate of 0.95. We found that the two references agree closely (inter-reference C2ST $\approx 0.55$), providing confidence that both recover the correct posterior.

The MNPE inference network consists of a categorical MADE for the discrete parameters and a neural spline flow (NSF) for the continuous parameters. Both components share the same width and depth: 4 hidden layers of 256 units. The NSF uses 4 coupling transforms with 9 rational-quadratic spline bins. Because queue length observations exhibit extreme right skew (median $\sim$0.1, tail extending to $\sim$10), we apply a $\log(1+x)$ transform to the two queue length dimensions before training. Training data is generated from the joint prior defined above, ensuring that the simulated data distribution matches the likelihood used for the MCMC reference.

## C.3 Hodgkin–Huxley simulator

The Hodgkin–Huxley simulator Hodgkin & Huxley (1952) action potential generation in neural tissue by considering the change in cell's membrane potential $V(t)$ in response to an external input $I_{\text{in}}(t)$ and how different voltage activated ion channels open and close in response to it.

We consider sodium, potassium and leak currents to generate spiking, a slow non-inactivating $K^+$ current that allows for spike-frequency adaptation, and low- (T-type) and high-threshold (L-type) $Ca^{2+}$ currents Pospischil et al. (2008):

$$
\begin{aligned}
C\,\frac{dV_t}{dt} =\,& I_t - g_{Na}m^3h(V_t - E_{Na}) - g_K n^4(V_t - E_K) - g_{leak}(V_t - E_{leak}) \\
& - g_M p(V_t - E_{Kt}) \\
& - g_{CaT}q^2 r(V_t - E_{Ca}) \\
& - g_{CaL}q^2(V_t - E_{Ca}) \\
& + \mathcal{N}(0, \sigma)
\end{aligned}
$$

with maximal conductances of the sodium, potassium, leak, adaptive potassium and calcium ion channels $g_i,\ i \in \{Na, K, leak, M, CaL, CaT\}$ and associated reversal potentials $E_i$. $C$ denotes the membrane capacitance, $I_t$ denotes the input current per unit area and $n, m, h, q, r$ and $p$ the fraction of opened channel gates.

The gating dynamics can be expressed as,

$$
\begin{aligned}
\frac{dz_t}{dt} &= \alpha_z(V_t)\,(1 - z_t) - \beta_z(V_t)z_t \\
\frac{dp_t}{dt} &= (p_\infty(V_t) - p_t)/\tau_p(V_t)
\end{aligned}
$$

with rate constants $\alpha_z(V_t)$ and $\beta_z(V_t)$ of $z \in \{m, n, h, q, r\}$ and $p$. For more detailed equations see Pospischil et al. (2008). The parameter values and bounds we used were adapted from (Pospischil et al., 2008) and Gonçalves et al. (2020) to cover biological meaningful ranges and are listed in Tab. C-1.

We used a step current $I_{inj}$ of $2\mu A/cm^2$ for $1000ms$ and run the simulation for $1450ms$. This stimulus and recording protocol corresponds to the voltage recordings from the Allen Institute for Brain Science (2016).

Table C-1: Parameter bounds and values that were used for our experiments.

| Parameter | Lower bound | Upper bound | Fixed Value | Observation Fig.4a (rounded) |
|---|---|---|---|---|
| $g_{Na}$ (mS) | 8 | 80 | - | 60.4 |
| $g_K$ (mS) | 1.5 | 15 | - | 3.9 |
| $E_{Na}$ (mV) | - | - | 50 | 50 |
| $E_K$ (mV) | - | - | -90 | -90 |
| $g_{leak}$ (mS) | 0.01 | 0.1 | - | 0.09 |
| $E_{leak}$ (mV) | -80 | -60 | - | -78.3 |
| $g_M$ (mS) | - | - | {0, 0.03} | 0.03 |
| $E_{Ca}$ (mV) | - | - | 120 | 120 |
| $g_{CaL}$ (mS) | - | - | {0, 0.1} | 0 |
| $g_{CaT}$ (mS) | - | - | {0, 0.4} | 0 |
| $\tau_p$ (s) | - | - | 1 | 1 |

Finally, we added independent Gaussian noise with a standard deviation of $\sigma = 0.1mV$ to each time point to simulate the stochasticity of the underlying biophysical processes.

## C.4 Implementation details

MNPE uses a neural spline flow with 5 coupling transforms and 8 blocks for the continuous parameters, and a MADE for the discrete parameters. Both networks have 6 hidden layers with 50 hidden features each. Training uses $N = 100, 500, 1000, ..., 100000$ simulations with a batch size of 500. We repeated each training with three distinct random seeds and averaged the evaluation results over these runs.

For the MSE evaluation, we used a test set of samples $(\boldsymbol{\theta}_o, \boldsymbol{x}_o)$, drawn from the joint $p(\boldsymbol{\theta})p(\boldsymbol{x}, \boldsymbol{\theta})$. We then draw a posterior sample $\boldsymbol{\theta}_{post} \sim p(\boldsymbol{\theta} \mid \boldsymbol{x}_o)$ and run a simulation $\boldsymbol{x}_{post} \sim p(\boldsymbol{x} \mid \boldsymbol{\theta}_{post})$. We then computed the MSE between $\boldsymbol{x}_o$ and $\boldsymbol{x}_{post}$ for each test sample. For the assessing the calibration of the continuous as well as the discrete parameters we used a test set of 10000 samples $(\boldsymbol{\theta}_o, \boldsymbol{x}_o)$, and draw 1000 posterior samples for continuous SBC per observation $\boldsymbol{x}_o$. For the discrete calibration we used 10 bins equidistantly tiling the interval $[0, 1]$.

