# OpenReview forum: "Mixed neural posterior estimation for simulators with discrete and continuous parameters"
_TMLR — Under review for TMLR_

### Review · Reviewer_1QqR · 2026-07-13

**Summary Of Contributions:**

- The paper introduces a simulation-based inference method to jointly model discrete-continuous parameter spaces. This works by factorizing the discrete and continuous parameter spaces, and then having two separate inference networks for them, in the discrete case outputting a class probability of the discrete parameters.
- The authors also introduce an empirical calibration test based on simulation-based calibration adapted to the discrete-continuous setting, which turns out to not be trivial in this extended setting.
- Finally, the authors test the method on four simulators of increasing complexity. The most complicated is a Hodgkin–Huxley neuroscience simulator, with the channel identities being the discrete parameters.

Overall this is a useful methodological extension to standard SBI methods that is likely to find use in scientific application. The main weakness I see is that the discrete-continuous parameter space problem is not solved by the method introduced in its full complexity. In particular, this includes discrete space of mixed cardinality and the transdimensional inference setting.

**Additional Comments:**

While the paper gracefully introduces SB-inference over discrete-continuous spaces, from my understanding it doesn't handle the case where we have objects of different cardinality (e.g., stars in a field with different properties, and the number of stars is unknown). This discrete-continuous case involves metamodels, i.e. unions of models with different cardinalities. In principle a fixed number of max discrete variables labeling objects can be introduced that can be turned on/off, but this introduced a host of other problems (like identifiable under permutation). In gist, I just wanted to mention this is an important use case not handled in the paper, and it would be great to discuss it upfront.

**Audience:**

Yes

**Audience Explanation:**

Yes, discrete-continuous parameter spaces are common in scientific models and I believe some TMLR readers (and the ML and science community more broadly) will find this methodological extension interesting. I have some comments about the overall scope, which I will place under "additional comments" since this question tests a specific bar.

**Broader Impact Concerns:**

No concerns.

**Claims And Evidence:**

Yes

**Claims Explanation:**

Overall the paper is clear and claims made are backed up by evidence. There are a couple places where I think either more evidence or more discussion could be helpful:
- The authors mention that an arbitrary number of discrete parameters with arbitrary class counts is supported. However, the number of discrete dimensions (not class counts ) is just a few max in all examples. Can the method reliably scale to higher discrete dimensions?
- The H-H example from what I understand doesn't have a ground truth baseline, so it's hard to verify the MNPE results for robustness (against the claim that the method works well on intractable likelihoods, which is this example).

**Requested Changes:**

- Primarily I would like to request the authors concretize the support behind the claims mentioned in the question above (critical)
- I would also request the authors account for the Additional Comments below when discussion scope, in particular talking about variable cardinality and transdimensional inference, as this is a critical setting of discrete-continuous inference that I believe is not supported by this method. This would help readers understand upfront the scope of the paper (e.g., before diving in I thought the paper does address the transdimensional inference question!)

---

### Review · Reviewer_2mSc · 2026-07-17

**Summary Of Contributions:**

This paper introduces Mixed Neural Posterior Estimation (MNPE), an extension of neural posterior estimation (NPE) to simulation-based inference problems with mixed discrete and continuous parameter spaces. The main contribution is a factorization of the posterior into a discrete component and a conditional continuous component:

$$p(\theta_d, \theta_c | x) = p(\theta_d | x) p(\theta_c | \theta_d, x),$$

where the discrete posterior is modeled using an autoregressive model and the continuous conditional posterior is modeled using  normalizing flows or diffusion models. This provides a modular framework that extends existing continuous NPE approaches to simulators containing categorical parameters.

The paper also proposes calibration diagnostics for mixed posteriors by combining simulation-based calibration (SBC) for continuous variables and classification calibration metrics (ECE/reliability diagrams) for discrete variables. The authors further introduce an empirical finite-sample baseline for interpreting discrete calibration errors.

The experimental evaluation covers three increasingly complex settings: (1) a tractable Gaussian simulator with an analytical posterior, (2) a queueing simulator where MCMC provides a reference posterior, and (3) a Hodgkin–Huxley simulator with an intractable likelihood. The results suggest that MNPE can recover accurate and calibrated mixed posteriors while retaining the amortization advantages of neural posterior estimation.

The main strengths of the work are:
- It addresses an important practical limitation of current neural posterior estimators, which typically assume continuous parameters.
- The proposed factorization is conceptually simple, probabilistically well motivated, and compatible with existing generative modeling techniques.
- The calibration discussion is valuable because mixed discrete-continuous posteriors require evaluation methods beyond standard continuous SBC.
- The experimental validation is reasonably comprehensive and includes both synthetic and scientific simulation problems.

The main weaknesses are:
- The core modeling idea is a natural combination of existing discrete classifiers and continuous density estimators, and the novelty compared with other hybrid generative modeling approaches could be clarified further.
- The scalability of the discrete component is not sufficiently evaluated for problems with many categorical variables or large discrete spaces.
- Comparisons with recent diffusion-based mixed inference approaches are mostly qualitative rather than empirical.
- The proposed calibration evaluation remains largely marginal and does not fully assess whether the dependency structure between discrete and continuous variables is correctly learned.

**Audience:**

Yes

**Audience Explanation:**

The findings are likely to be of interest to a substantial portion of the TMLR audience.

The paper addresses a general problem in probabilistic inference: learning posterior distributions when the latent parameter space contains both discrete and continuous variables. This setting appears broadly in scientific machine learning, Bayesian inference, probabilistic programming, and generative modeling.

Researchers working on simulation-based inference, neural density estimation, conditional generative models, and Bayesian deep learning would likely find the proposed framework relevant. The modular design also makes the approach potentially useful to practitioners who already use normalizing flows, diffusion models, or other conditional generative models for inference.

Beyond SBI, the idea of combining autoregressive categorical models with continuous posterior generators may also be relevant to researchers studying hybrid discrete-continuous generative models.

Although the paper targets a relatively specialized application area, the underlying modeling problem is sufficiently general that the findings should be of interest to the broader TMLR readership.

**Broader Impact Concerns:**

I do not identify significant ethical concerns associated with this work.

**Claims And Evidence:**

Yes

**Claims Explanation:**

The main claims of the paper are generally supported by the experimental results presented.

The authors demonstrate that MNPE can recover known posterior distributions in the tractable Gaussian example, where the method successfully captures multimodal continuous posteriors induced by uncertainty over discrete variables. The comparison against MCMC references in the queueing simulator provides evidence that the method can approximate mixed posteriors accurately in a setting where likelihood-based inference is available. The Hodgkin–Huxley experiment further demonstrates applicability to a realistic simulator with an intractable likelihood.

The calibration experiments are also a strength. The paper correctly recognizes that standard SBC does not directly apply to discrete posterior components and provides a practical combination of continuous rank-based calibration and discrete classification calibration. The proposed finite-sample baseline improves the interpretation of ECE values.

However, some claims would benefit from additional evidence. In particular, claims regarding scalability and advantages over alternative mixed inference approaches are not fully demonstrated experimentally. The discrete component is only tested on relatively small categorical spaces, and the comparison with diffusion-based mixed inference methods remains conceptual. Additional experiments would strengthen these claims.

Overall, the evidence is convincing for the demonstrated problem settings, but broader empirical validation would be needed to establish MNPE as a generally scalable solution for high-dimensional mixed parameter spaces.

**Requested Changes:**

The following changes would strengthen the paper:

## Major suggestions

1. Compare against alternative mixed inference methods.

The paper discusses diffusion-based approaches but does not provide empirical comparisons. Experiments comparing MNPE with representative diffusion-based mixed inference methods would strengthen the claims regarding computational efficiency, posterior quality, and scalability.

2. Improve evaluation of joint posterior structure.

The current calibration analysis focuses mainly on marginal dimensions. However, the key contribution of MNPE is modeling the dependency:

p(\theta_c|\theta_d, x).

The authors should include additional diagnostics evaluating whether the learned coupling between discrete and continuous variables is correct. Possible approaches include conditional SBC, posterior predictive checks conditioned on discrete states, or dependence-based metrics.

## Minor suggestions

- Include an ablation comparing the proposed conditional factorization against an independent approximation:
  p(theta_d|x)p(theta_c|x).
  This would quantify the benefit of explicitly modeling discrete-continuous dependencies.

- Investigate sensitivity to the ordering of discrete variables in the autoregressive factorization.

- Provide more detailed runtime comparisons between MNPE inference and MCMC baselines, since amortized inference speed is an important motivation.

- Discuss more clearly when explicit enumeration over discrete states is feasible and when MNPE provides substantial advantages.

---

### Review · Reviewer_qgD2 · 2026-07-17

**Summary Of Contributions:**

The paper tackles the problem of inferring mixed discrete and continuous parameters in simulation-based inference. The authors introduce MNPE, which extends NPE to handle both parameter types jointly via a factorized posterior, implemented using a MADE for the discrete component and a standard continuous generative model for the conditional continuous component.


### Strengths
- Extending NPE to handle both discrete and continuous parametric spaces is novel and interesting.
- The modular design is clean and practical, allowing any continuous generative model as a drop-in backbone and enabling integration with existing libraries such as sbi.

**Audience:**

Yes

**Audience Explanation:**

The paper tackles a practically important problem, mixed discrete–continuous parameter inference in simulation-based inference, that arises naturally across many scientific domains. The modular design, which allows any continuous generative model as a drop-in backbone, lowers the barrier to adoption and makes the method easy to integrate into existing SBI workflows.

**Claims And Evidence:**

Yes

**Claims Explanation:**

The core empirical claims are well-supported across three simulators of increasing complexity, with MNPE recovering analytical posteriors in the Gaussian case, matching MCMC on the queueing simulator, and producing calibrated posteriors on Hodgkin–Huxley. The use of C2ST, SBC, and ECE as complementary metrics further strengthens the credibility of these claims.

**Requested Changes:**

### Questions
1. The factorization in Section 3.1 separates the training objective into two independent terms, implying the two subnetworks could be trained independently. Is there a specific reason joint training was chosen, and do the authors have an ablation comparing joint versus separate training?
2. Also, variational dequantization has been used to handle discrete structure within continuous generative models in molecule, protein, and text domains. Did the authors consider applying standard NPE to dequantized discrete parameters, and see how it performs as a baseline?
3. Is the factorization choice $p(θ_d | x) · p(θ_c | θ_d, x)$ motivated by domain considerations or modeling convenience? Did the authors investigate whether the reverse factorization $p(θ_c | x) · p(θ_d | θ_c, x)$ affects posterior quality or training stability?
4. Since the continuous network is conditioned on $θ_d$ at inference time, how sensitive is the joint posterior quality to errors in the discrete posterior estimate? Is there an analysis of how miscalibration in the discrete component propagates to the continuous posterior?
5. Could the authors provide a concrete inference algorithm or pseudocode clarifying how $θ_d$ samples are passed as conditioning inputs to the continuous network at test time, as this sequential dependency is not clearly described in the main text or appendix?
6. The empirical ECE baseline depends on the bin count B, yet B = 10 is used throughout without sensitivity analysis. Could the authors report how ECE values and baselines vary across different choices of B, or justify why B = 10 is appropriate across all settings?